# Structural insights into flagellar stator–rotor interactions

**Yunjie Chang[1,2†], Ki Hwan Moon[1,3†‡], Xiaowei Zhao[2,4§], Steven J Norris[4], MD A Motaleb[3*], Jun Liu[1,2,4*]**

[1]Department of Microbial Pathogenesis, Yale University School of Medicine, New Haven, United States; [2]Microbial Sciences Institute, Yale University, West Haven, United States; [3]Department of Microbiology and Immunology, Brody School of Medicine, East Carolina University, Greenville, United States; [4]Department of Pathology and Laboratory Medicine, McGovern Medical School at University of Texas Health Science Center at Houston, Houston, United States

**Abstract** The bacterial flagellar motor is a molecular machine that can rotate the flagellar filament at high speed. The rotation is generated by the stator–rotor interaction, coupled with an ion flux through the torque-generating stator. Here we employed cryo-electron tomography to visualize the intact flagellar motor in the Lyme disease spirochete, *Borrelia burgdorferi*. By analyzing the motor structures of wild-type and stator-deletion mutants, we not only localized the stator complex in situ, but also revealed the stator–rotor interaction at an unprecedented detail. Importantly, the stator–rotor interaction induces a conformational change in the flagella C-ring. Given our observation that a non-motile mutant, in which proton flux is blocked, cannot generate the similar conformational change, we propose that the proton-driven torque is responsible for the conformational change required for flagellar rotation.

DOI: https://doi.org/10.7554/eLife.48979.001

**\*For correspondence:**
MOTALEBM@ecu.edu (MDAM);
jliu@yale.edu (JL)

[†]These authors contributed equally to this work

**Present address:** [‡]Division of Marine Bioscience, Korea Maritime and Ocean University, Busan, Republic of Korea; [§]HHMI Janelia Research Campus, Ashburn, United States

**Competing interests:** The authors declare that no competing interests exist.

## Introduction

Many bacterial pathogens require motility to infect, disseminate, and cause disease in humans and other mammalian hosts. Among diverse motility machineries, the flagellum is the best understood among bacteria. The flagellum consists of a motor, hook, and long filament (*Macnab, 2003*; *Terashima et al., 2008*; *Berg, 2003*). The motor is a sophisticated nanomachine composed of a rotor, which is the rotary part, and a stator, which surrounds the rotor. The rotation of the motor is driven by the interaction between the rotor and the stator, which is powered by the proton or sodium gradient across the cytoplasmic membrane (*Berg, 2003*; *Sowa and Berry, 2008*; *Minamino et al., 2008*). However, how the ion gradient couples the mechanical rotation remains elusive at the molecular level.

The stator has been extensively characterized in *Escherichia coli, Salmonella enterica*, and several other species (*Berg, 2003*; *Macnab, 2003*; *Terashima et al., 2008Beeby et al., 2016*). Each stator complex is composed of two transmembrane proteins: MotA and MotB (*Braun and Blair, 2001*; *Kojima and Blair, 2004a*; *Sato and Homma, 2000*). MotA has a large cytoplasmic domain, which contains conserved charged residues that are critical for the interaction with the rotor-associated protein FliG (*Zhou and Blair, 1997*). MotB has a large periplasmic domain that is believed to bind to the peptidoglycan layer (*Chun and Parkinson, 1988*). However, stator complexes in some motor are constantly exchanged with those in the membrane pool (*Leake et al., 2006*), resulting in the dynamic nature and variable assembly of the stator complexes (*Block and Berg, 1984*; *Blair and Berg, 1988*; *Reid et al., 2006*). The assembly of the stator is mediated by ion flow in *E. coli* and *Vibrio alginolyticus* (*Tipping et al., 2013*; *Fukuoka et al., 2009*), as its disruption leads to reversible

stator complex diffusion away from the motor. A conserved aspartic acid residue in the transmembrane segment of MotB (D32 in *E. coli* and D33 in *S. enterica*) is the predicted proton-binding site and plays a crucial role for torque generation and bacterial motility (*Zhou et al., 1998a*). It is thought that proton binding/dissociation at this residue triggers conformational changes in the cytoplasmic domain of MotA to produce a power-stroke on the C-ring that drives flagellar rotation (*Kojima and Blair, 2001*).

Many electron microscopy techniques have been deployed to visualize the stator complexes. Electron micrographs of freeze-fractured membrane showed the stator complexes as stud-like particles in *E. coli*, *Streptococcus*, and *Aquaspirillum serpens* (*Khan et al., 1988*; *Coulton and Murray, 1978*). Recently, electron microscopy of purified PomA/PomB (a MotA/MotB homolog) revealed two arm-like periplasmic domains and a large cytoplasmic domain (*Yonekura et al., 2011*). However, the isolated stator unit without its context is insufficient for determining how stator complexes are arranged in an intact flagellar motor. Cryo-electron tomography (cryo-ET) has recently emerged as an advanced approach for visualizing the intact flagellar motor in a bacterial envelope (*Chen et al., 2011*; *Murphy et al., 2006*; *Kudryashev et al., 2010*; *Liu et al., 2009*; *Beeby et al., 2016*). However, detailed visualization of stator units remains challenging in many bacterial species including *E. coli* and *S. enterica*, because the stator complexes are highly dynamic and wild type cells are often too large for high resolution cryo-ET imaging. In contrast, stator complexes have been observed in many intact flagellar motors, including spirochetal flagellar motors (*Chen et al., 2011*; *Murphy et al., 2006*; *Kudryashev et al., 2010*; *Liu et al., 2009*; *Liu et al., 2010a*; *Zhao et al., 2014*) and polar flagellar motors (*Beeby et al., 2016*; *Zhu et al., 2017b*). Presumably, the stators in these motors either are not exchanged, or exchange occurs without disrupting the overall arrangement of the stator units. Nevertheless, previous structure images of spirochetal flagellar motors were relatively low in resolution, and thus insufficient for identifying the stator complexes and their interaction with other motor components in detail.

*Borrelia burgdorferi*, one of the agents of *Lyme* disease, is the model system for understanding unique aspects of spirochetes. The flagella in *B. burgdorferi* are enclosed between the outer membrane and the peptidoglycan layer and are thus called periplasmic flagella. The flagellar motors are found at both cell poles and rotate coordinately to enable the cell to run, pause, or flex (*Charon et al., 2012*). The rotations of periplasmic flagella cause the cell to form a flat-wave shape to efficiently bore its way through tissue and viscous environments (*Charon et al., 2012*; *Moriarty et al., 2008*; *Motaleb et al., 2000*; *Motaleb et al., 2015*). Although the *B. burgdorferi* motor differs in some aspects from the *E. coli* motor (e.g. the presence of a prominent 'collar' structure), genome sequence analyses as well as in situ structural analyses suggest that the major flagellar components are remarkably similar to those of other bacterial species (*Qin et al., 2018*; *Zhao et al., 2014*). Studies of many flagellar rod mutants have yielded insights concerning rod assembly (*Zhao et al., 2013*). Moreover, studies of *fliI* and *fliH* mutants have provided structural information about the export apparatus and its role in the assembly of periplasmic flagella (*Lin et al., 2015*; *Qin et al., 2018*). Furthermore, membrane protein FliL was identified as an important player that controls periplasmic flagellar orientation (*Motaleb et al., 2011*). More recently, the collar (a spirochete-specific feature) was determined to play an important role in recruiting and stabilizing the stator complexes (*Moon et al., 2016*; *Moon et al., 2018*). Thus, *B. burgdorferi* is a tractable model organism for studying the structure and function of bacterial flagellar motors at molecular resolution (*Zhao et al., 2014*).

Here, we focused primarily on determining the structure and function of the stator complex in *B. burgdorferi*. By comparative analysis of the motor structures from wild-type, stator mutants, and complemented strains, we localized the stator complexes in the spirochetal flagellar motor. Importantly, detailed analysis of the stator–rotor interaction revealed a novel conformational change that is necessary for transmitting torque from the stator to the C-ring.

## Results

### MotA/MotB complex in *B. burgdorferi* is the torque-generating unit powered by proton gradient

Previous studies have shown that wild-type (WT) *B. burgdorferi* cells were immobilized after being treated with proton uncoupler Carbonyl cyanide 3-chlorophenylhydrazone (CCCP), indicating that the proton gradient is used for flagellar rotation (*Motaleb et al., 2000*). As Δ*motB* cells were completely non-motile (*Sultan et al., 2015*), and both *motA* and *motB* genes are located in the middle of the *flgB* operon in *B. burgdorferi* (*Figure 1A*), the MotA/MotB complex is the torque generating unit essential for spirochetal motility, though not for flagellar assembly (*Figure 1B*). We constructed deletion mutants Δ*motB* and Δ*motA*, showing that both mutant cells are non-motile and have irregular and rod-shaped morphology − very different from the highly motile wave-like WT or complemented *motB*com cells (*Figure 1C*, *Figure 1—figure supplement 1*. Both Δ*motA* and Δ*motB* cells possess paralyzed flagella, indicating that both mutants lack the torque-generating unit (*Figure 1D*), which is consistent with the notion that MotA and MotB form a stator complex necessary for torque generation (*Kojima and Blair, 2004b*).

Aspartic acid in MotB (*B. burgdorferi* Asp-24, *E. coli* Asp-32, and *S. enterica* Asp-33) is highly conserved (*Figure 1—figure supplement 2*), and it is thought to be directly involved in proton translocation in *E. coli* and *S. enterica* (*Zhou et al., 1998a*; *Che et al., 2008*). To confirm its specific role in *B. burgdorferi*, we introduced point mutations to generate *motB*-D24E and *motB*-D24N. Dark-field microscopy and swarm plate motility assays indicated that *motB*-D24E mutant cells are less motile than the WT cells whereas *motB*-D24N cells are completely non-motile (*Figure 1E,F*). The result obtained with *motB*-D24E is consistent with the reduced motility observed with the D32E substitution in *E. coli* MotB (*Zhou et al., 1998a*) and the D33E substitution in *S. enterica* (*Che et al., 2008*). The non-motile phenotype in *B. burgdorferi motB*-D24N is also identical to that of the D32N substitution in *E. coli* MotB (*Zhou et al., 1998a*; *Blair et al., 1991*). Therefore, we concluded that the MotA/MotB complex is the torque-generating unit in *B. burgdorferi*, and Asp-24 in MotB is essential for proton translocation.

### Characterization and localization of the torque-generating units in intact *B. burgdorferi* motor

To image the stator complexes and their interactions with other flagellar components in situ at the molecular level, we generated asymmetric reconstructions of the flagellar motor from Δ*motB* and Δ*motA* strains (*Figure 2*, *Figure 2—figure supplement 1*, *Supplementary file 1*). The averaged structures of the motors in the Δ*motB* and Δ*motA* mutants exhibited many of the same features as the WT motor, such as the export apparatus, the C-ring, the MS-ring, the rod, the P-ring, and the spirochete-specific periplasmic collar. However, juxtaposed with the WT motor structure, both Δ*motA* and Δ*motB* mutants lacked large transmembrane densities peripheral to the C-ring (indicated by arrows in *Figure 2*, *Figure 2—figure supplement 1*, *Video 1*). Complementation of Δ*motB* restored the missing densities of the Δ*motB* mutant (*Figure 2—figure supplement 1*) and also restored WT motility and cell morphology (*Figure 1—figure supplement 1*). These results suggest that the peripheral densities in the WT motor are comprised of MotA/MotB complexes. In each WT motor, sixteen of these densities are symmetrically distributed around the C-ring (*Figure 2E*). They form a stator ring with 16-fold symmetry and ~80 nm diameter, which is significantly larger than the Δ*motB* mutant's C-ring diameter of ~57 nm (*Figure 2E*, *Video 1*).

### The stator–rotor interaction induces conformational changes in the C-ring

Compared to the C-ring in the Δ*motB* mutant, the bottom portion of the WT C-ring has the same diameter of 56 nm, while the top portion undergoes considerable changes in the presence of the stator complexes. Specifically, the diameter of the C-ring reduces from 59 nm in the Δ*motB* to 57 nm in the WT (*Figure 2B and E*). As the motor is embedded in the cytoplasmic membrane, both C-ring and collar are apparently modulated by the curved membrane in both Δ*motB* (*Figure 3A–C*) and WT motors (*Figure 3D–F*). Although the 16-fold symmetry of the collar is well preserved (*Figure 3B,C*), the symmetry of the C-ring is not obvious in the Δ*motB* mutant, indicating that there

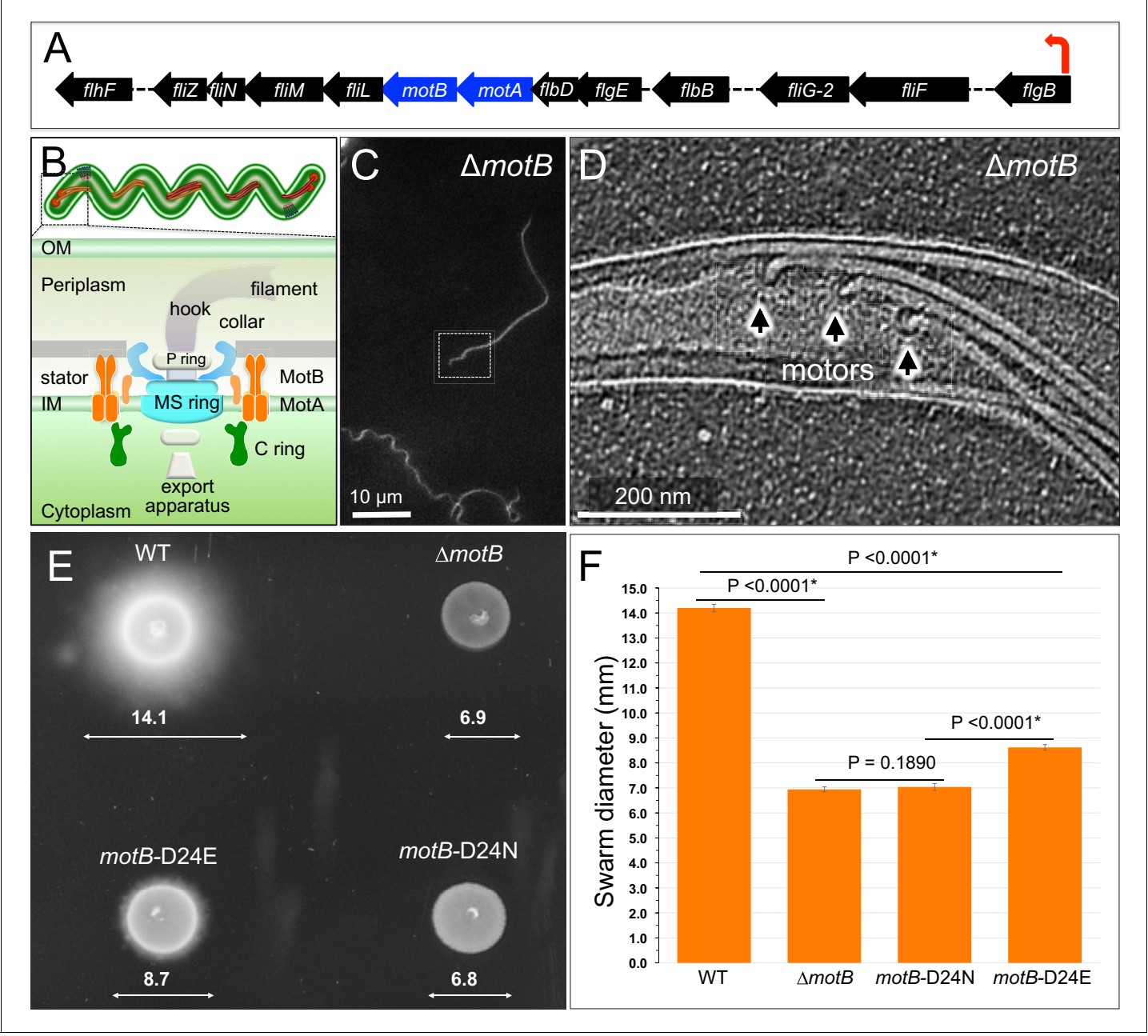

**Figure 1.** Overview of flagellar organization in *Borrelia burgdorferi* and the motility phenotypes of WT, Δ*motB*, and point mutants of *motB*. (**A**) Schematic of the *flgB* flagellar operon map of *B. burgdorferi*. Red arrow indicates the direction of transcription. The *motA* and *motB* genes are shown as blue arrows. (**B**) Schematic models of the periplasmic flagellum and the motor in a spirochete cell. (**C**) A dark-field microscopy image of a Δ*motB* cell. (**D**) A section from a typical tomogram of a Δ*motB* cell tip shows multiple flagellar motors and filaments *in situ*. (**E**) Swarm plate assay of WT, Δ*motB*, *motB*-D24E, and *motB*-D24N cells. (**F**) Averages ± standard deviations of swarm diameters from WT, Δ*motB*, *motB*-D24E and *motB*-D24N strains. A paired Student's *t* test was used to determine a *P* value. P<0.05 between strains is considered significant.

DOI: https://doi.org/10.7554/eLife.48979.002

The following figure supplements are available for figure 1:

**Figure supplement 1.** Construction and characterization of Δ*motB* and *motB* complementation.
DOI: https://doi.org/10.7554/eLife.48979.003

**Figure supplement 2.** Sequence alignment of MotB proteins from three bacteria: *B. burgdorferi* (B.), *E. coli* (E.) and *S. enterica* (S.).
DOI: https://doi.org/10.7554/eLife.48979.004

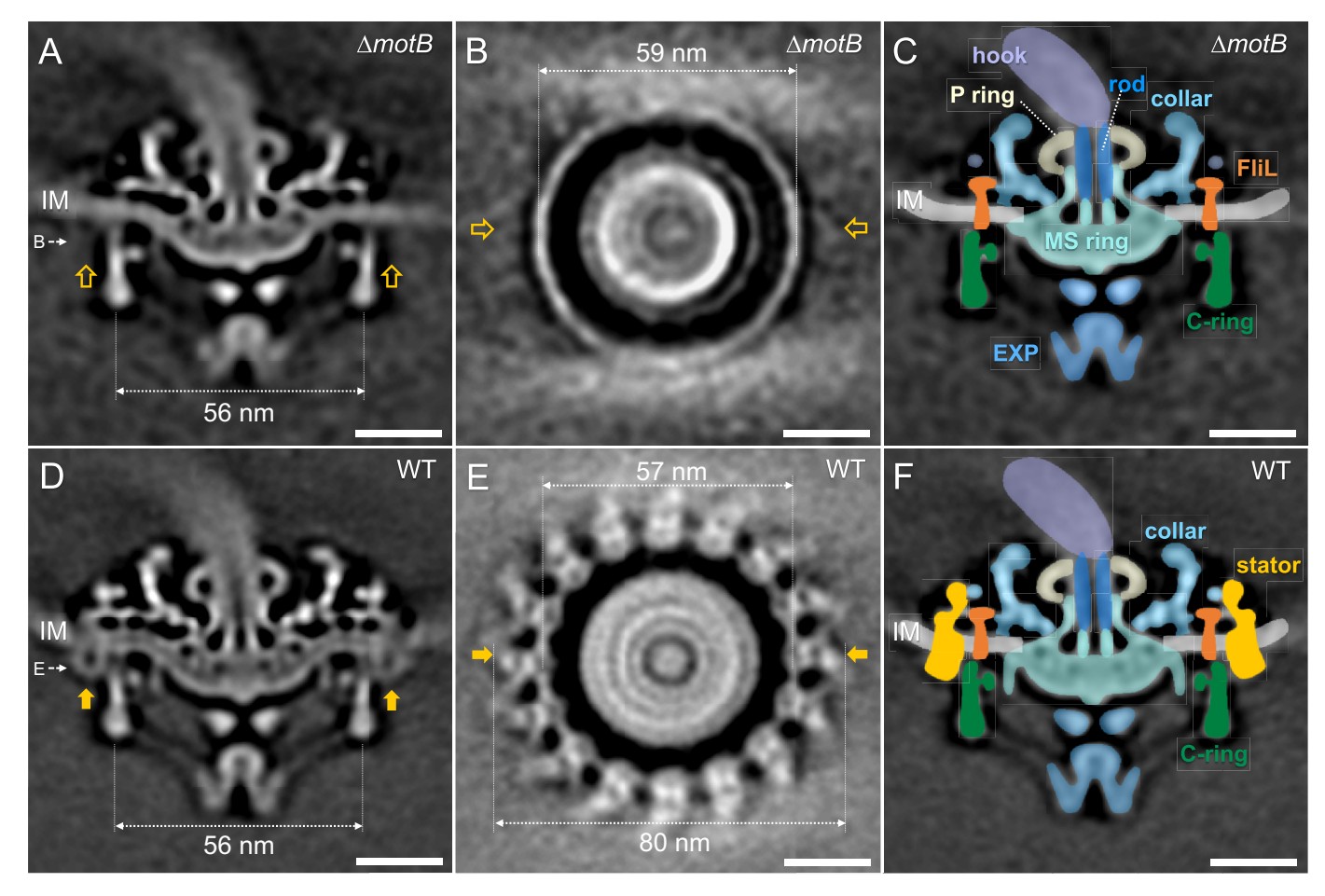

**Figure 2.** Asymmetric reconstructions of the Δ*motB* and WT motors in *B. burgdorferi*. (**A**) A central section of the flagellar motor structure from a Δ*motB* mutant. The diameter of the bottom of the C-ring is 56 nm. The missing densities compared to the WT flagellar motor are indicated by empty yellow arrows. (**B**) A cross-section at the top of the C-ring from the Δ*motB* flagellar motor structure. The diameter of the top of the C-ring is 59 nm. (**C**) A cartoon model highlights key components in the Δ*motB* flagellar motor: C-ring (green), export apparatus (EXP), MS-ring (blue-green) embedded in the inner membrane (IM), FliL (coral), collar (light blue), P-ring (gray), and rod (blue). (**D**) A central section of the flagellar motor structure from WT. The diameter of the bottom of the C-ring is 56 nm. The extra densities compared to Δ*motB* flagellar motor structure are indicated by solid orange arrows. (**E**) A cross-section at the top of the C-ring from the WT flagellar motor structure. The diameter of the top of C-ring is 57 nm. Note that there are sixteen stator densities associated with the C-ring. The diameter of the stator ring is 80 nm. (**F**) A cartoon model highlights key flagellar components in the WT flagellar motor: C-ring, MS-ring, FliL, collar, and stators. Scale bar = 20 nm.

DOI: https://doi.org/10.7554/eLife.48979.005

The following figure supplement is available for figure 2:

**Figure supplement 1.** Asymmetric reconstructions of Δ*motA*, Δ*motB* and *motB*[+] motors in *B. burgdorferi*.

DOI: https://doi.org/10.7554/eLife.48979.006

a mismatch between the C-ring and the collar. Indeed, a combination of asymmetric reconstruction and focused alignment allows us to show for the first time that the C-ring in the WT motor has a 46-fold symmetry (*Figure 3E* and *Figure 3—figure supplement 1*), while the surrounding stator ring and the collar have a 16-fold symmetry (*Figure 3E,F*). Importantly, the sixteen stator complexes in the WT motor also display considerable variation in height as they are embedded in the cytoplasmic membrane and their periplasmic domains are inserted between the collar (*Figure 3H*).

To obtain a high-resolution structure of the stator complex and its interaction with the C-ring, the sixteen stator complexes from every motor were rotationally aligned and classified. Class averages of Δ*motB* and WT motors embedded in the flat membrane (as shown in *Figure 3G–J*) were selected for further comparative analysis and in situ structure determination of the stator complex (*Figure 3I*

**Video 1.** Conformational change of the C-ring from the ΔmotB motor to the WT motor.
DOI: https://doi.org/10.7554/eLife.48979.007

and J). The stator complex is composed of a periplasmic portion and a cytoplasmic portion (*Figure 3I and J*). The periplasmic portion is ~14 nm in height, and it directly interacts with the collar and FliL (*Figure 3I*), which have been previously identified as periplasmic structures (*Motaleb et al., 2011*). The interactions are believed to play critical roles in stabilizing the stator, as the lack of FliL or collar proteins has a profound impact on stator assembly and motility (*Motaleb et al., 2011*; *Moon et al., 2016*; *Moon et al., 2018*). Notably, the collar-FliL assembly also exhibits conformational changes due to the absence or presence of the stator complexes (*Figure 3H,J*), further confirming protein-protein interactions among the stator complex and the collar (*Figure 3J*).

The cytoplasmic portion of the stator complex is ~8 nm in diameter, which is comparable to the structure observed in the freeze-fractured membrane (*Khan et al., 1988*; *Coulton and Murray, 1978*; *Khan et al., 1992*), and is adjacent to the top portion of the C-ring (*Figure 3I*). Importantly, the interaction between the stator and the C-ring induces a noticeable conformational change in the C-ring (*Figure 3I*), compared to that in the ΔmotB mutant motor (*Figure 3G*). The top of the C-ring appears to be tilted toward the MS-ring by ~6° in the presence of the stator complex (*Figure 3I*). Importantly, this local conformational change is consistent with the overall diameter change of the C-ring (*Figure 2B,E*) observed in ΔmotB and WT motors.

## Structural impacts of the proton gradient

To gain a better understanding of the conformational changes observed in the C-ring in the presence of the torque-generating stator, we examined the motor structures of the less-motile *motB*-D24E mutant and the non-motile *motB*-D24N mutant, and compared them with the motor structures of WT and ΔmotB mutants (*Figure 4*). Both the *motB*-D24E and *motB*-D24N mutants have stator complexes assembled in the motor (*Figure 4A–D*), but the stator densities are significantly weaker than that in WT cells (by comparing *Figure 4A–D* with *Figure 2D*), suggesting that the stator complexes vary in their location or occupancy. Statistical analysis of the stator complexes (details described in the Materials and Methods section) shows that stator occupancy is 97.0% in WT cells, suggesting that there are ~16 stator complexes in each WT motor. As a control, the ΔmotB mutant had no observed stator complex. In the *motB*-D24E motor, stator occupancy is ~62.5%, suggesting that there are ~10 stator complexes in each motor on average. In *motB*-D24N, it is estimated that there are ~7 stator complexes in each motor on average (*Figure 4* and *Figure 4—figure supplement 1*). As these two residue substitutions are known to respectively reduce or block proton flux in the stator complex, reduced stator occupancy in these mutants is consistent with the notion that the proton gradient affects stator assembly and stability (*Tipping et al., 2013*; *Fukuoka et al., 2009*).

Importantly, compared to the ΔmotB motor's C-ring (*Figure 4E*), that in the *motB*-D24N motor undergoes a minor change in the presence of the stator complex (*Figure 4F*). The tilt angles of the C-ring in the ΔmotB motor (7.8° in *Figure 4E*) and the *motB*-D24N motor (6.6° in *Figure 4F*) are considerably different from that in the WT motor (1.8° in *Figure 4H*). As the proton channel is likely blocked in the *motB*-D24N motor, no torque is generated by the stator complex. As a result, the C-ring conformation in the *motB*-D24N is similar to that in the ΔmotB motor. To further examine the impact of the proton gradient on the C-ring, we analyzed the nonmotile WT motors after 15 min of treatment with CCCP. All 16 stator complexes remained associated with the motor (*Figure 4—figure supplement 1*). However, the tilt angle of the C-ring was 5.1°, which is different from that in the motile WT motor (*Figure 4—figure supplement 2*). Therefore, the conformational change of the C-ring likely results from the torque generated by the proton gradient.

*MotB*-D24E cells are less motile than WT cells (*Figure 1E*). There is a noticeable difference in tilt angles of the C-ring between the *motB*-D24E motor (3.2° *Figure 4G*) and the ΔmotB motor (7.8° in *Figure 4E*). As the residue substitution D24E was known to reduce proton flux, we expected there

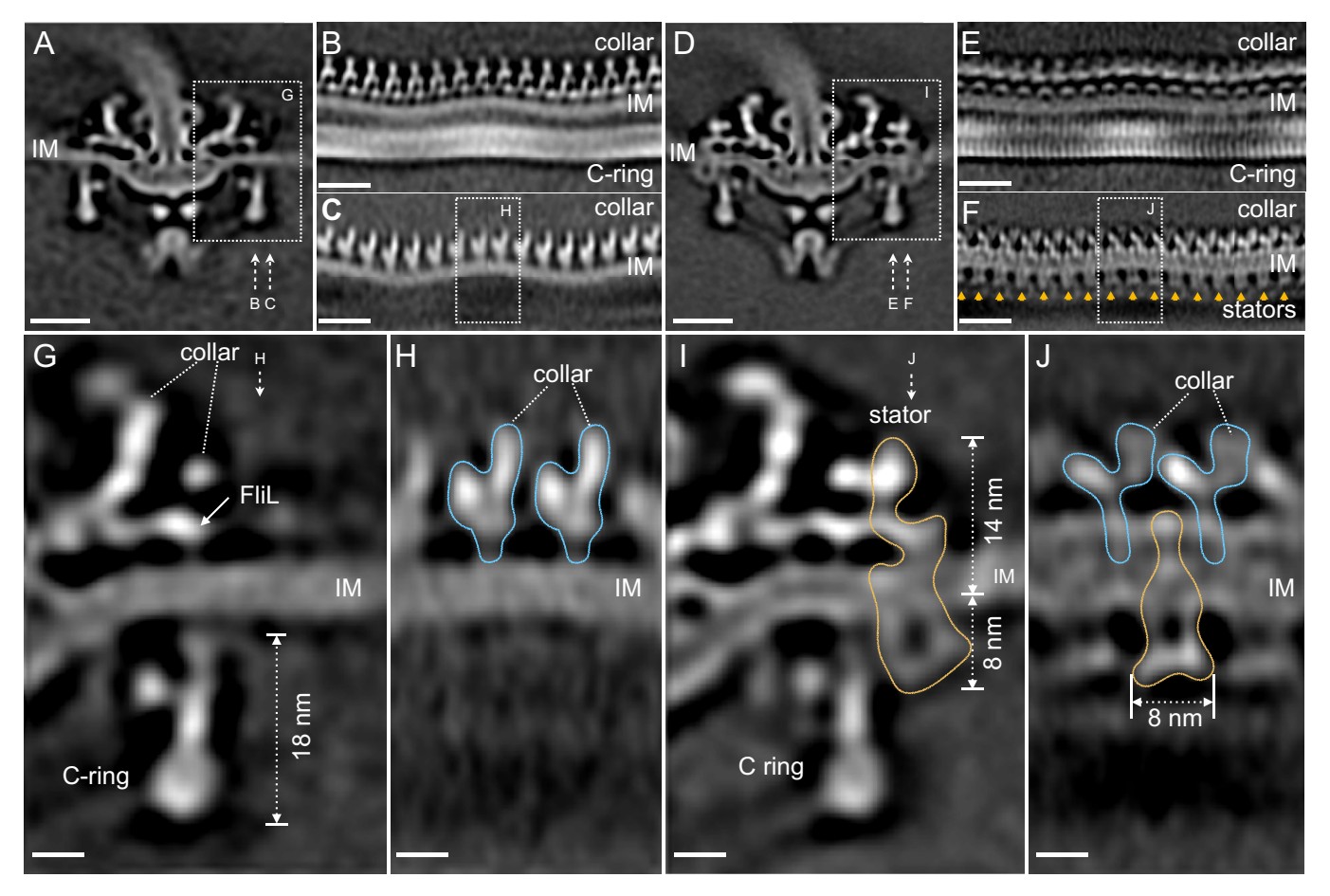

**Figure 3.** Visualization of the stator complexes and their interactions with the flagellar periplasmic components and the C-ring. (A) A central section of the ΔmotB motor structure. (B, C) Two sections from an unrolled map of the ΔmotB motor showing the curvature of the inner membrane (IM), the collar and the C-ring, respectively. (D) A central section of the WT motor structure. (E) One section from the unrolled map of the refined C-ring of the WT motor showing the symmetry mismatch between the C-ring and the collar. (F) Another section from the unrolled map of the WT motor (D) showing sixteen stator complexes (indicated by orange arrows) embedded in the IM. (G) A central section from a refined structure of the ΔmotB motor showing the C-ring, collar and FliL embedded in the IM. (H) A perpendicular section showing the collar (blue line highlighted) on top of the IM. (I) A central section from a refined structure of the WT motor showing the C-ring, and the stator complex (gold line). (J) A perpendicular section showing the stator complex (gold line) inserted between two subunits of the collar (highlighted by blue lines). Scale bar in panels A-F is 20 nm. Scale bar in panels G-J is 20 nm.

DOI: https://doi.org/10.7554/eLife.48979.008

The following figure supplement is available for figure 3:

**Figure supplement 1.** Symmetry analysis of the stator and the C-ring in WT motor structure.

DOI: https://doi.org/10.7554/eLife.48979.009

to be less torque generated by the stator complex. Although we were not able to directly measure the torque, we provide evidence that there was less conformational change in the *motB*-D24E motor when compared to WT cells. Altogether, our data indicate that the conformational change of the C-ring correlated with the proton flux transmitting through the stator channel.

## Architecture of stator–rotor interface in *B. burgdorferi*

To model the stator–rotor interactions and the conformational change of the C-ring in detail, we built pseudo-atomic structures of the C-ring in the ΔmotB mutant. The FliN tetramer was placed into the bulge at the bottom of the C-ring, as proposed for *E. coli* (*Brown et al., 2005*). The FliM_M-FliG_MC complex (PDB: 4FHR) (*Vartanian et al., 2012*) and the N-terminal domain of FliG (PDB 3HJL)

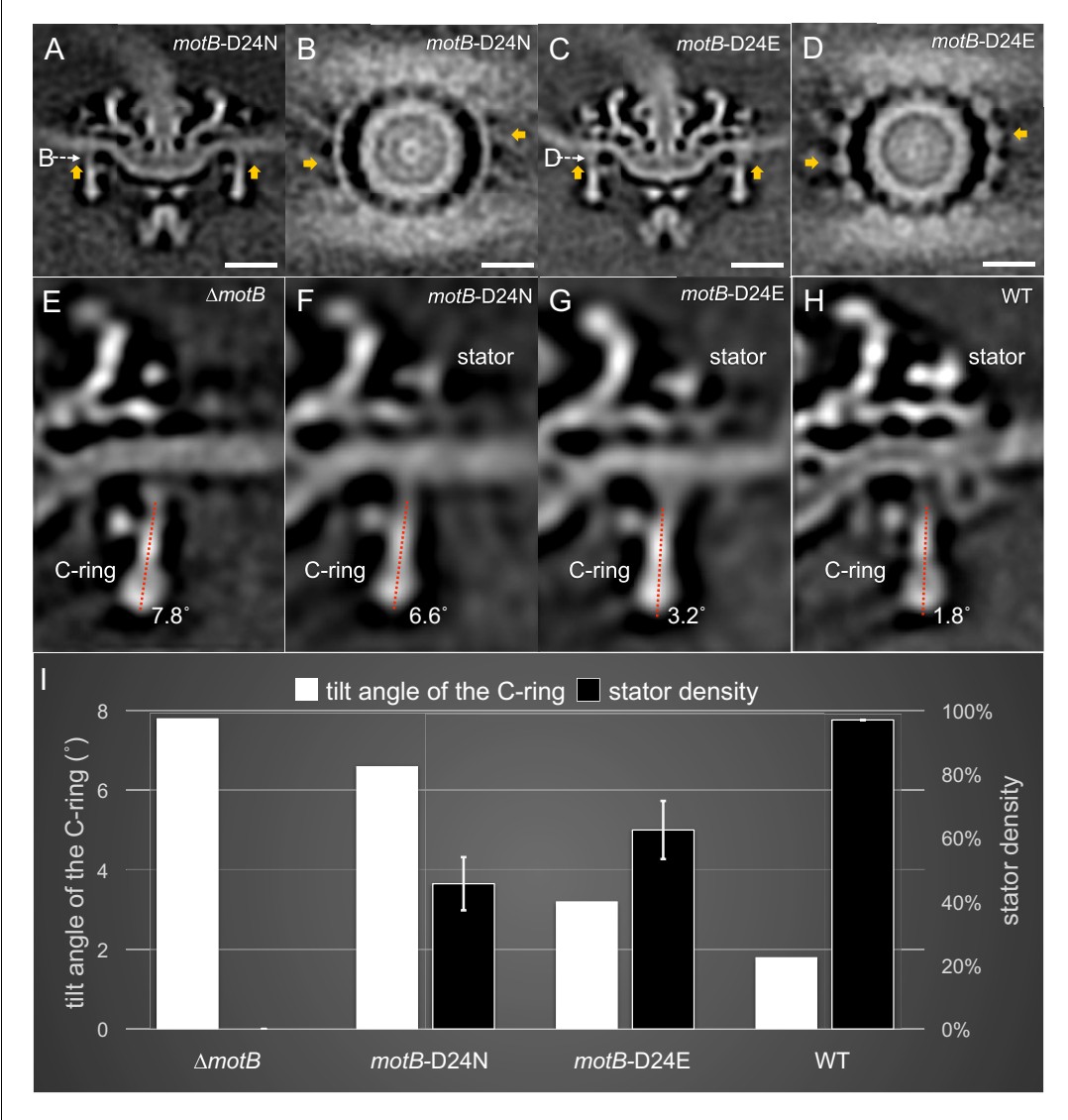

**Figure 4.** Stator binding and proton flux are required to induce the conformational changes of the C-ring. (**A**) A central section from the asymmetric reconstruction of the motB-D24N motor. (**B**) A cross-section from the asymmetric reconstruction of the motB-D24N motor at the interface between the C-ring and the stator. (**C**) A central section from the asymmetric reconstruction of the motB-D24E motor. (**D**) A cross-section from the asymmetric reconstruction of the motB-D24E motor at the interface between the C-ring and the stator. Two stator units are indicated by arrows in (**A-D**). Note that the stator densities in the motB-D24E and motB-D24N mutants are considerably weaker than that in the WT motor. For a better comparison, we showed the central sections from the refined ΔmotB motor structure (**E**), the refined motB-D24N motor structure (**F**), the refined motB-D24E motor structure (**G**) and the refined WT motor structure (**H**), respectively. The tilt angle of the C-ring away from the rotation axis is about 7.8˚ in the ΔmotB motor (**E**), while those in the motB-D24N, motB-D24E and WT motors are 6.6˚, 3.2˚, 1.8˚, respectively. (**I**) 3D classification based on the C-ring density and stator complex density reveals various conformations of the C-ring and different occupancy of the stator units in four strains: WT, motB-D24E, motB-D24N, and ΔmotB. Scale bar is 20 nm.

DOI: https://doi.org/10.7554/eLife.48979.010

The following figure supplements are available for figure 4:

**Figure supplement 1.** Measurement of stator occupancies in WT, motB-D24E, motB-D24N, ΔfliL, and CCCP treated WT motors.

DOI: https://doi.org/10.7554/eLife.48979.011

**Figure supplement 2.** Conformational changes of the C-ring induced by stator-binding and proton flux.

DOI: https://doi.org/10.7554/eLife.48979.012

**Figure supplement 3.** Comparison of the motor structures from WT, ΔfliL and ΔmotB.

DOI: https://doi.org/10.7554/eLife.48979.014

**Figure supplement 4.** Measurement of the C-ring tilt angle in the ΔmotB motor structure.

*Figure 4 continued on next page*

*Figure 4 continued*

DOI: https://doi.org/10.7554/eLife.48979.013

(*Lee et al., 2010*) were docked onto the top portion of the C-ring. The densities were well-fitted with 46 FliG proteins organized in a ring in our EM map (*Figure 5*, *Video 2*). Noticeably, FliG proteins are relatively far away from the periphery of the MS-ring, and the C-ring appears to be disengaged from the MS-ring. In the WT motor, the FliG/FliN/FliM complex has to be rotated for about ~6° to fit into the C-ring density (*Figure 5*, *Video 2*). The additional shifts of the FliG proteins enables the C-ring to engage with the MS-ring at its periphery. As a result, the N-terminal domain of FliG interacts with the periphery of the MS-ring, and the C-terminal domain of FliG interacts with the stator. It has been demonstrated that several charged residues in FliG and MotA are important for torque generation in *E. coli* (*Zhou and Blair, 1997*; *Zhou et al., 1998b*; *Lloyd et al., 1999*). Consistent with these findings, our model shows that two charged residues in the C-terminal domain of FliG are adjacent to the cytoplasmic portion of the stator complexes, which presumably interact with the charged residues of MotA (*Figure 5*). Powered by the proton gradient, the stator–rotor interactions induce a large conformational change of the C-ring, consequently driving flagellar rotation.

## Discussion

The bacterial flagellum is arguably one of the most fascinating bacterial motility organelles. It has been studied extensively in *E. coli* and *S. enterica* model systems for several decades. However, our

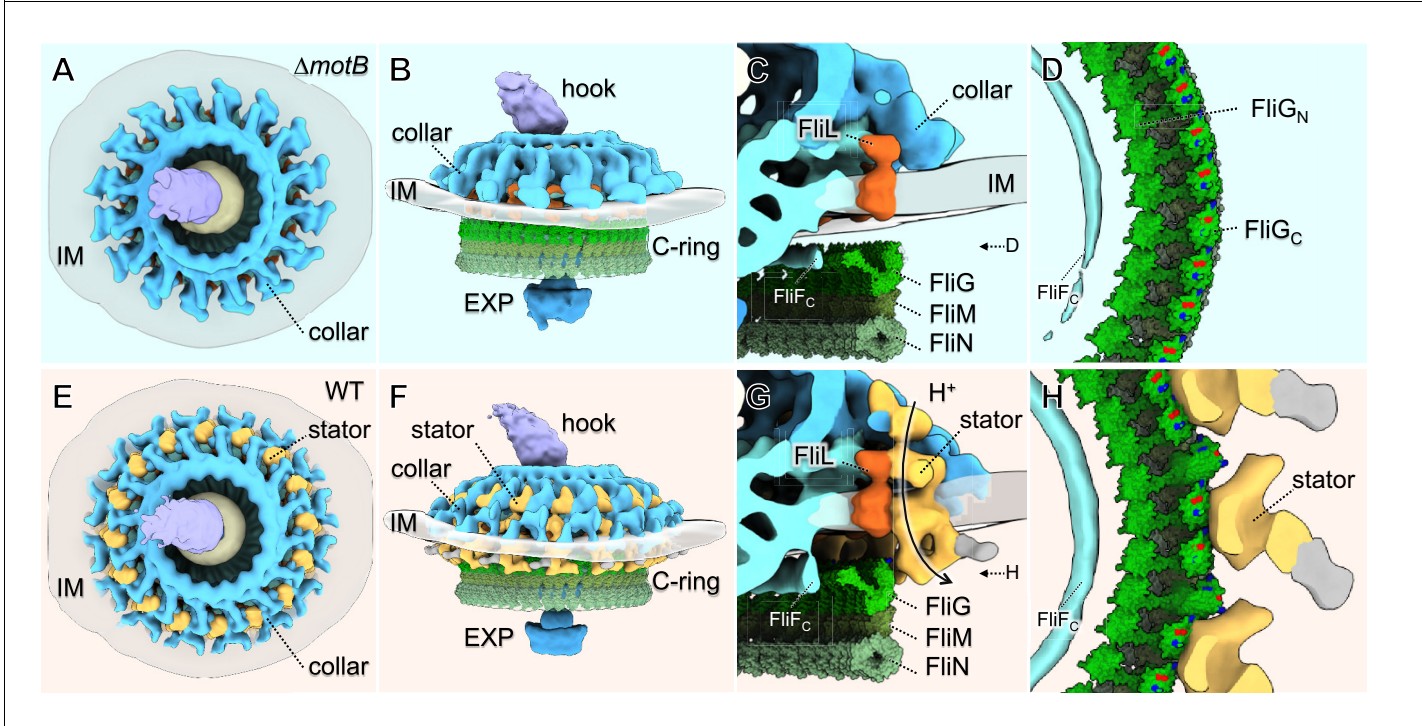

**Figure 5.** Molecular architecture of the stator-rotor interactions. (A, B) A top and a side view showing the surface rendering of the ΔmotB motor, respectively. (C) A zoom-in view shows major flagellar components: the cytoplasmic domain of FliF (FliFC), FliL, FliG, FliM, FliN and the collar around the inner membrane (IM). (D) A top view of the interface between the C-ring and the MS-ring. FliFC of the MS-ring is adjacent to the FliGN of the C-ring. (E, F) A top and a side view showing the surface rendering of the WT motor. (G) A side view of the interface between the stator and the C-ring. The interaction powered by proton flux induces a conformational change of the C-ring, which appears to engage with the MS-ring through interactions between FliGN and FliFC. (H) A top view of the interface between the C-ring and the stators. The charged residues in FilGC are shown in blue (positive electrostatic potential) or red (negative electrostatic potential).
DOI: https://doi.org/10.7554/eLife.48979.015

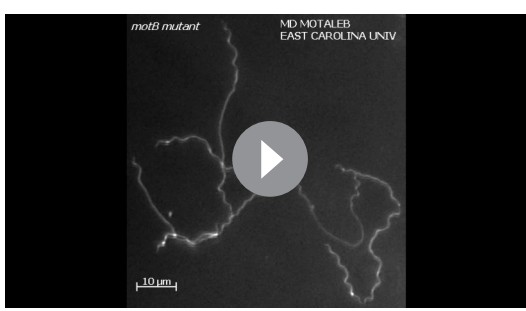

**Video 2.** Visualization of the stator-rotor interactions in *B. burgdorferi*.
DOI: https://doi.org/10.7554/eLife.48979.016

understanding of flagellar assembly and rotation remains incomplete, partly because the torque-generating stator is highly dynamic, and structural information about the stator complex and its interaction with the rotor is limited. Recent studies have provided clear evidence that flagella from different bacterial species evolved considerably and variably in flagellar structure, number, and placement to adapt to the specific environments that bacteria encounter. Spirochetes are unique in their evolution of periplasmic flagella, resulting in a form of locomotion effective in viscous environments such as host tissues. Spirochetal flagellar motors not only are significantly larger than those found in *E. coli* or *S. enterica*, but also possess a unique periplasmic collar structure. In this study, the relative stability and high occupancy rate of the *B. burgdorferi* stator, together with the availability of key mutants, permitted the use of cryo-ET analysis to reveal the structure of the stator complex and its interactions with the rotor at high resolution, and thus provided new insights into the mechanisms underlying flagellar motor assembly and rotation.

## The unique structure of periplasmic flagella is critical to stator assembly

The periplasmic collar constitutes a large, turbine-like complex in the flagellar motor of spirochetes. This structure plays an important role in the assembly of periplasmic flagella, and hence, in determining cell morphology and motility (*Moon et al., 2016*; *Moon et al., 2018*). In contrast to the highly dynamic stator complexes in the *E. coli* flagellar motor, sixteen stator complexes in *B. burgdorferi* appear to be stably assembled around the collar (*Figure 2* and *Figure 3*). In the absence of the collar, the stator complexes were no longer visible in sub-tomogram averages, suggesting that the collar is important for assembling and stabilizing the stator in *B. burgdorferi* (*Moon et al., 2016*; *Moon et al., 2018*). In addition, FliL forms additional periplasmic structures between the stator and the collar (*Motaleb et al., 2011*). Deletion of *fliL* resulted in cells with defects in motility (*Motaleb et al., 2011*) and a flagellar motor with fewer stator units (*Figure 4—figure supplement 1*, *Figure 4—figure supplement 3*), suggesting that FliL also plays an important role in stator assembly in *B. burgdorferi*. Together, both the collar and FliL in *B. burgdorferi* enable us to visualize the stator assembly and its impacts on the C-ring.

## Impact of the proton gradient on stator and C-ring assembly in periplasmic flagella

The proton gradient is not only essential for flagellar rotation but also critical for assembly of the stators around the motor in *E. coli* and other model systems. By altering the putative proton channel in *B. burgdorferi*, we found that the average number of stator units decreased significantly in the less motile *motB*-D24E (65% occupancy) and non-motile *motB*-D24N (45% occupancy) motors (*Figure 4*). However, even in the non-motile *motB*-D24N cells, some stator units remained associated with the motor. This finding is different from the observations in *E. coli* and other model systems, in which stator units were found to dissociate from both Na$^+$- and H$^+$-driven motors when the ion motive force was disrupted (*Tipping et al., 2013*; *Fukuoka et al., 2009*). The periplasmic collar in spirochetes may be the key reason underlying the difference, as deletion of genes that encode the proteins of the collar also disrupt the assembly of the stator units (*Moon et al., 2016*; *Moon et al., 2018*).

It has been proposed that proton flow through the motor triggers conformational changes in the stator that generate a power stroke to the C-ring (*Kojima and Blair, 2001*). However, it has been difficult to directly observe the conformational changes in the stator complex, partly because of the dynamic nature of the stator and its interactions with the C-ring. To visualize the conformational changes in detail, we took advantage of several unique features in the *B. burgdorferi* flagellar system: 1) the large periplasmic collar and FliL help to recruit sixteen stator units to each motor; 2) a

higher torque is presumably required to drive the rotation of periplasmic flagella and the entire cell body; 3) multiple motors located at the skinny poles enable high-resolution cryoET imaging; 4) recent advances in the genetic tools available for *B. burgdorferi* enable specific mutations. As a result, we were able to observe a large C-ring conformational change resulting from stator assembly, proton transport, and torque generation. Importantly, by comparing the structures of Δ*motB*, *motB*-D24N, and *motB*-D24E (*Figure 4* and *Figure 4—figure supplement 2*), we found that the conformational change of the C-ring from *motB*-D24N to *motB*-D24E (conformational change of the top portion and 3.4° tilt angle change) is much more significant than that from Δ*motB* to *motB*-D24N (no change of the top portion and only 1.2° tilt angle change). This indicates that stator-binding alone is not sufficient to induce the large C-ring conformational changes because stator complexes associated with the *motB*-D24N motor are not able to interact effectively with the C-ring and drive flagellar rotation, as proton conduction is blocked in the *motB*-D24N motor. Therefore, we conclude that the torque induced by proton flux is required for the C-ring conformational changes and flagellar rotation.

## Stator–rotor interaction and its impacts on flagellar C-ring and rotation

Our in situ structural analysis of *B. burgdorferi* flagellar motors enables us to propose a model for stator assembly and stator–rotor interactions. Before assembly of the stator complexes, the *B. burgdorferi* flagellar motor is composed of the MS-ring, the rod, the export apparatus, the collar with 16-fold symmetry, and the C-ring with 46-fold symmetry (*Figure 6*). The collar and the associated FliL protein provide 16 well-defined locations for the recruitment of stator complexes, which assemble around the collar and the C-ring. Stator complexes with blocked proton conduction only partially occupy the 16 possible locations (*Figure 6*). There is little conformational change in the C-ring at the stator–rotor interface. As proton conduction increases, more stator positions become occupied. Larger conformational changes of the C-ring occur with higher torque generated by stator complexes with increasing proton-conduction capability (*Figure 6*).

In summary, high resolution in situ structural analysis of the flagellar motor from wild-type and mutant in *B. burgdorferi* has provided new insights into the assembly of torque-generating stators and their interactions with other flagellar components in both the periplasm and cytoplasm. Coupled with the proton gradient, stator–rotor interactions trigger large conformational changes required for flagellar rotation.

# Materials and methods

## Bacterial strain and growth conditions

High-passage, avirulent *B. burgdorferi* sensu stricto strain B31A and its isogenic mutants (*Supplementary file 1*) were grown in Barbour-Stoenner-Kelly medium without gelatin (BSK II) or plating BSK medium containing 0.5% agarose at 35°C in a 2.5% $CO_2$ humidified incubator.

## Construction of Δ*motA*, Δ*motB*, *motB*D24E, *motB*D24N, and complementation of *motB*

Constructions of Δ*motB* (gene locus number *bb0280*) and Δ*motA* (locus number *bb0281*) were described previously (*Sultan et al., 2015*). Shuttle vector pBSV2G, which carries the gentamicin cassette, was used to complement the *motB::aadA* mutant (Δ*motB*) using native *motB* promoter, *flgB*. The *flgB* promoter and *motB* gene sequences were amplified with primers containing restriction enzyme sites *Xba*I and *Nde*I (5'−3' and 3'−5' respectively) and inserted into the *Nde*I site (the 3' end of the promoter fragment and the 5' end of the *motB* gene) to yield pFlgBMotB. The primers used were (5'−3'): for *flgB*, FlgB-XbaI-F: tctagagccggctaatacccgagc and FlgB-NdeI-R: <u>catatg</u>gaaacctccct-catttaa; and for *motB*, MotB.com-F: <u>catatg</u>gctttgcgaattaaga and MotB.com-R: <u>tctaga</u>ttactgct-taatttcctt. Underlined sequences indicate restriction sites. The *flgBmotB* DNA was then ligated into the *Xba*I site of pBSV2G to yield pMotB.com. Two point mutations were generated in MotB (aspartate to glutamate and asparagine, respectively) as follows (*Pazy et al., 2010*). We used pMotB.com plasmid as the PCR template for these substitutions using a site-directed mutagenesis kit (Quik-Change, Stratagene Inc) yielding plasmids MotB-D24E and MotB-D24N, respectively. These

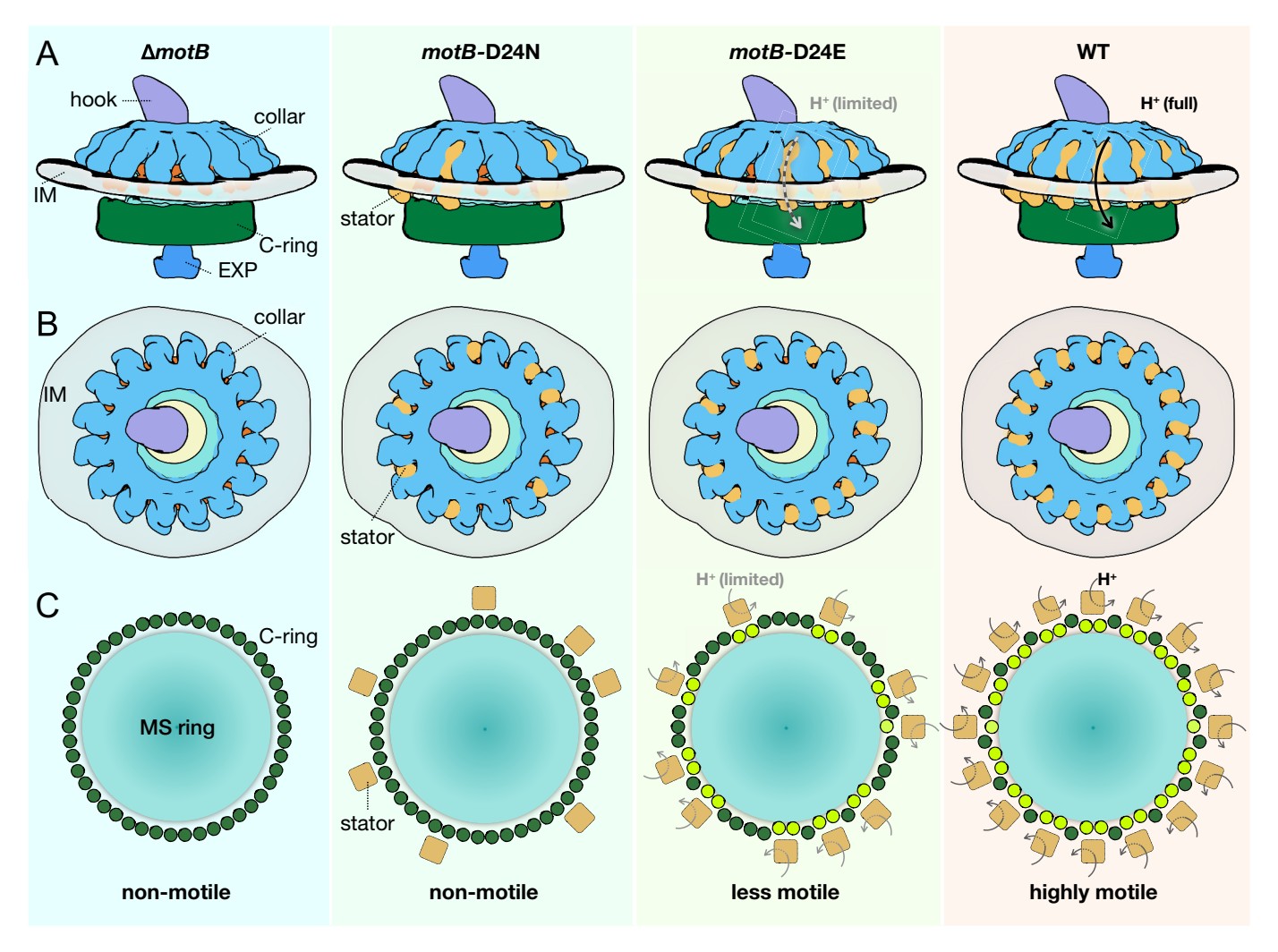

**Figure 6.** Schematic of stator assembly and stator-rotor interactions. (**A, B**) Side and top views of stator assembly. A flagellar motor without stators is shown in the left panels. The WT motor with 16 fully assembled stators is shown in the right panels. Several key flagellar components are annotated: C-ring (green), export apparatus (EXP), collar (light blue) embedded in the inner membrane (IM), and hook. Two intermediates in stator assembly show partial occupancy of motors with blocked or attenuated proton conduction by the torque-generating units. (**C**) Top views of the C-rings from the motors shown in panels A and B. Without proton conduction (as shown in the motB-D24Nmutant), some stators (tan squares) interact with the FliG units (colored in dark green circles), yet there is relatively little conformational change in the C-ring. When protons flow through the stator channels, the torque generated by the stator induces conformational changes in the FliG units with which they are in contact (colored in light green). In the motB-D24E motor, fewer stator units are engaged, and they have a decreased proton flow. As a consequence, the deformations of the C-ring are not as large as in the WT motor, in which the 16 stator units assembled around the C-ring are rapidly conducting protons and generating torque. We propose that the increasing deformation of the C-ring observed with increasing number and activity of assembled stator complexes reflects conformational changes induced by the power strokes of the cytoplasmic MotA loops pushing against the FliGc domain.

DOI: https://doi.org/10.7554/eLife.48979.017

plasmids were sequenced to verify the substitutions. PCR primer sequences for point mutations are given below (5′−3′), and underlined sequences indicate point mutated sequences:

P11 – gttgacttatggagaaatggttactttgctg

P12 – cagcaaagtaaccatttctccataagtcaac

P13 – gttgacttatggaaatatggttactttgctg

P14 – cagcaaagtaaccatatttccataagtcaac.

Approximately 20 µg of purified pMotB.com, MotB-D24E, and MotB-D24N plasmids were used to transform competent ΔmotB by electroporation as described above. Transformants were selected

with 40 µg/ml gentamicin plus 80 µg/ml streptomycin. Gentamicin and streptomycin resistant clones were confirmed by PCR as well as by western blotting to determine the restoration of MotB synthesis.

## Gel electrophoresis and western blot analysis

Sodium dodecyl sulfate-polyacrylamide gel electrophoresis (SDS-PAGE) and western blotting with an enhanced chemiluminescent detection method were carried out as described previously (*Sultan et al., 2013*).

## Dark-field microscopy and swarm plate motility assays

B.*B. burgdorferi* cells ($5 \times 10^7$ spirochetes/ml) were observed under a dark-field microscope (Zeiss Axio Imager M1), and images were captured using an AxioCam digital camera. Swarm plate motility assays were performed as described (*Sultan et al., 2015*). Approximately $1 \times 10^7$ cells in a 5 µl volume were spotted onto 0.35% agarose plate containing plating BSK medium diluted 1:10 in Dulbecco's phosphate buffered saline. Since *B. burgdorferi* is a slow growing organism (8–12 hr generation time), plates were incubated for 5 days at 35˚C in a 2.5% $CO_2$ humidified incubator. To determine cell morphology, growing *B. burgdorferi* cells were observed under a dark-field microscope (Zeiss Axio Imager. M1).

## Cryo-EM sample preparation

Cultured WT and mutant cells were centrifuged in 1.5 ml tubes at $5,000 \times$ g for 5 min and the resulting pellet was rinsed gently with PBS, then, suspended in 40 µl PBS at a final concentration ~$2 \times 10^9$ cells/ml (*Liu et al., 2009*). To prepare CCCP treated WT cells, CCCP was added to a final concentration of 20 µM as described previously (*Motaleb et al., 2000*). After incubating for 15 min, cell motility was observed to make sure that the cells were non-motile. After mixing with 10 nm gold fiducial markers, 5 µl *B. burgdorferi* samples were deposited onto freshly glow-discharged holey carbon grids. Grids were blotted with filter paper for ~3–5 s and then rapidly frozen in liquid ethane, using a home-made gravity-driven plunger apparatus as described previously (*Liu et al., 2009*; *Zhao et al., 2013*).

## Cryo-electron tomography

The frozen-hydrated specimens were transferred to a 300-kV Polara G2 electron microscope (FEI) equipped with a Direct Electron Detector (DDD) (Gatan K2 Summit) or with a charge-coupled-device (CCD) camera (TVIPS; GMBH, Germany). Images were recorded at 15,400 × magnification with pixel size of 2.5 Å (for images recorded by K2) or at 31,000 × magnification with pixel size of 5.7 Å (for images recorded by CCD). SerialEM (*Mastronarde, 2005*) was used to collect tilt series at −6 to −8 µm defocus, with a cumulative dose of ~100 e⁻/Å (*Terashima et al., 2008*) distributed over 61 images and covering angles from −60˚to 60˚, with a tilt step of 2˚. Images recorded by K2 camera were first drift-corrected using the *motioncorr* program (*Li et al., 2013*). Then all tilt series were aligned using fiducial markers by IMOD (*Kremer et al., 1996*), tilt images were contrast transfer function corrected using 'ctfphaseflip' function in IMOD, and tomograms were reconstructed by weighted back-projection using TOMO3D (*Agulleiro and Fernandez, 2015*).

## Subtomogram averaging and correspondence analysis

Bacterial flagellar motors were manually picked from the tomograms as described (*Zhu et al., 2017a*). The subtomograms of flagellar motors were extracted from the bin1 tomograms first, then binning by 2 or four based on the requirement for alignment. In total, 14,049 subtomograms were manually selected from the tomographic reconstructions and used for subtomogram analysis. The i3 software package (*Winkler, 2007*; *Winkler et al., 2009*) was used for subtomogram analysis including alignment and classification. Class averages were computed in Fourier space so that the missing wedge problem of tomography was minimized (*Winkler et al., 2009*; *Liu et al., 2010b*). Fourier shell correlation coefficients were calculated by generating the correlation between two randomly divided halves of the aligned images used to estimate the resolution and to generate the final maps.

To identify symmetry of the C-ring in the WT motor, focused alignment and classification were applied for the C-ring after getting initial asymmetric reconstruction of the entire WT motor. During

the processing, a molecular mask around the C-ring was applied to the reference, and the angular search range along the motor rod was restricted to be smaller than 2° so that we can maintain overall alignment of the motor.

Focused alignment and classification were used to analyze the rotor-stator interactions. The densities around the sixteen stator units in each motor were first extracted and aligned, then 3D classification was applied based on the stator complex, the C-ring and the cytoplasmic membrane features. Only those containing flat membrane were selected for further analysis. For *motB*-D24N, *motB*-D24E and WT motors, those particles that do not have stator complex density were not used for further refinement. Number of subtomograms used for focused alignment were listed in *Supplementary file 2*.

To objectively measure the stator occupancy, regions around the sixteen stator units in each motor were first aligned, then 3D classification was applied based on the stator complex density. Basically, three different kinds of class averages can be obtained: class average with stator complex density, class average without stator complex density and class average that we are not sure whether there is stator complex or not (examples are shown in *Figure 4—figure supplement 1*). Then the stator occupancy was calculated by dividing the particle number in the class average with stator complex density by the total particle number.

To objectively estimate the tilt angle of the C-ring, 16-fold symmetry was first applied to the motor structure. A cross section of the motor structure was selected as shown in *Figure 4—figure supplement 4*. An ellipse was generated to fit the C-ring density (without $FliG_N$ density). The angle between the long axis of the ellipse and the Y-axis was considered as the tilt angle of the C-ring.

## Three-dimensional visualization and modeling

UCSF Chimera (*Pettersen et al., 2004*) and ChimeraX (*Goddard et al., 2018*) software packages were used for surface rendering of subtomogram averages and molecular modeling. Unroll maps of the motor structures were generated using 'vop unroll' function of UCSF Chimera (*Pettersen et al., 2004*). For the surface rending of the WT motor structure, all stator densities from the focused alignment are shown in *Figure 3I*, then fitted into the motor density shown in *Figure 2D* through the function 'fitmap' in Chimera or ChimeraX, thus the 16 stator complexes are almost the same. They have relatively different orientations and positions. For the surface rendering of Δ*motB* motor structure, the density map shown in *Figure 2A* was used. The crystal structures of $FliG_N$ (PDB ID: 5TDY), $FliM_C$-$FliG_{MC}$ complex (PDB ID: 3HJL) and FliN (PDB ID: 1YAB) (*Brown et al., 2005*) were docked into the density map through the function 'fitmap' in Chimera.

## Acknowledgements

We thank Mike Manson for critical reading and suggestions. We also thank Jonathan Sigworth for proofreading the manuscript. This research was supported by grants R01AI087946 (to JL) and R01AI132818 (to MM) from the National Institute of Allergy and Infectious Diseases and R01GM107629 from the National Institute of General Medicine (to JL).

## Additional information

### Funding

| Funder | Grant reference number | Author |
|---|---|---|
| National Institute of Allergy and Infectious Diseases | R01AI087946 | Jun Liu |
| National Institute of Allergy and Infectious Diseases | R01AI132818 | MD A Motaleb |
| National Institute of General Medical Sciences | R01GM107629 | Jun Liu |

The funders had no role in study design, data collection and interpretation, or the decision to submit the work for publication.

## Author contributions

Yunjie Chang, Data curation, Investigation, Visualization, Writing—original draft, Writing—review and editing; Ki Hwan Moon, Data curation, Investigation; Xiaowei Zhao, Data curation, Formal analysis, Investigation, Visualization, Writing—original draft; Steven J Norris, Conceptualization, Investigation, Writing—review and editing; MD A Motaleb, Conceptualization, Data curation, Supervision, Investigation, Writing—original draft, Writing—review and editing; Jun Liu, Conceptualization, Data curation, Supervision, Funding acquisition, Investigation, Methodology, Writing—original draft, Writing—review and editing

## Author ORCIDs

Jun Liu (iD) https://orcid.org/0000-0003-3108-6735

## Decision letter and Author response

Decision letter https://doi.org/10.7554/eLife.48979.030
Author response https://doi.org/10.7554/eLife.48979.031

## Additional files

### Supplementary files

• Supplementary file 1. Strains used in this study.
DOI: https://doi.org/10.7554/eLife.48979.018

• Supplementary file 2. Cryo-ET data used in this study.
DOI: https://doi.org/10.7554/eLife.48979.019

• Transparent reporting form DOI: https://doi.org/10.7554/eLife.48979.020

### Data availability

Data have been placed in the Electron Microscopy Data Bank under the accession numbers EMD-0534, EMD-0536, EMD-0537, and EMD-0538.

The following datasets were generated:

| Author(s) | Year | Dataset title | Dataset URL | Database and Identifier |
|---|---|---|---|---|
| Chang Y, Moon KH, Zhao X, Norris SJ, Motaleb MA, Liu J | 2019 | Asymmetric reconstruction of the in situ flagellar motor structure in Borrelia burgdorferi | https://www.ebi.ac.uk/pdbe/entry/emdb/EMD-0534 | Electron Microscopy Data Bank, EMD-0534 |
| Chang Y, Moon KH, Zhao X, Norris SJ, Motaleb MA, Liu J | 2019 | Local refinement of stator-rotor interaction region in flagellar motor of wild type Borrelia burgdorferi | https://www.ebi.ac.uk/pdbe/entry/emdb/EMD-0536 | Electron Microscopy Data Bank, EMD-0536 |
| Chang Y, Moon KH, Zhao X, Norris SJ, Motaleb MA, Liu J | 2019 | cryo-ET flagellar motor structure of motB deletion Borrelia burgdorferi | https://www.ebi.ac.uk/pdbe/entry/emdb/EMD-0537 | Electron Microscopy Data Bank, EMD-0537 |
| Chang Y, Moon KH, Zhao X, Norris SJ, Motaleb MA, Liu J | 2019 | Local refinement for the in-situ flagellar motor structure of motB deletion Borrelia burgdorferi | https://www.ebi.ac.uk/pdbe/entry/emdb/EMD-0538 | Electron Microscopy Data Bank, EMD-0538 |

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
