## [Decision Letter]

[Editors’ note: a previous version of this study was rejected after peer review, but the authors submitted for reconsideration. The first decision letter after peer review is shown below.]

Thank you for submitting your work entitled "In situ structures of periplasmic flagella in *Borrelia* reveals conformational changes essential for flagellar rotation" for consideration by *eLife*. Your article has been reviewed by two peer reviewers, and the evaluation has been overseen by a Reviewing Editor and a Senior Editor.

Our decision has been reached after consultation between the reviewers. Based on these discussions and the individual reviews below, we regret to inform you that your work will not be considered further for publication in *eLife*.

The reviewers were all in agreement that the paper was very impressive from a technical point of view. However, the main unique conclusion of the paper, involving ion flow, was seen as speculative. It is quite possible that a substantially revised paper including actual data on ion flow would surmount the current concerns, but *eLife* does not offer such major revision decisions. If the authors can provide such a paper in the future we would be willing to consider it further.

*Reviewer #1:*

This paper observed the periplasmic flagellar motor of *Borrelia burgdorferi* with electron cryo-tomography (ECT).It compared the ECT images of the flagella motor of the wild type and its derivative mutants, and then focused on both the number of stators incorporated by the flagellar motor and the tilt angle of the C-ring away from the rotation axis. It is a very interesting study. The results of each experiment are also solid. The descriptive parts of the results and the discussion are thoughtfully considered.

Reviewer #2:

This interesting paper describes use of *Borrelia burgdorferi* as a model to study conformational changes during torque generation by the bacterial flagellar motor. The authors are particularly interested in the stator complex motor proteins, and image wild-type motors and compare them with two MotB mutants, D24E (reduced motility) and D24N (abolished motility), and a motB¬-deletion. Comparison of these structures reveals conformational changes in the C-ring-primarily a change in C-ring angle-that the authors suggest will provide clues to future understanding the mechanism of flagellar motor torque generation.

The paper is well-written and technically well executed (indeed, is a technical tour-de-force in some ways), although has unclear biological significance. I have a number of comments:

Major comments:

While the results are noteworthy as technical accomplishments, it is unclear whether the "conformational change" is an important observation with implications for understanding torque generation, or whether it is simply that the C-ring is no longer held in place by the stator complexes. A possibly related observation of the C-ring conformation changing upon stator engagement has already been observed (in *Campylobacter jejuni*, Beeby and Hendrixson labs, Beeby et al., 2016), and interpreted as C-ring flexibility when not engaged by the stator complexes. Can the authors argue against this null hypothesis? If the authors do have reason to believe that this is a mechanistically-relevant conformation change as opposed to increased flexibility in the absence of the stator complexes, this would be a significant result. If not, the proposed "conformational changes" may not tell us much about torque generation.

Use of the word "essential" in the title is inaccurate: nothing in the results reveals that these conformational changes are essential for flagellar rotation.

The Materials and methods section, and descriptions of procedures used in the Results section, is entirely inadequate, and needs considerable elaboration, as follows.

– How did the authors identify 46-fold symmetry? What software did the authors use? Was the C-ring averaged separately to the stator ring, and the two subsequently merged? If not, how were both symmetries resolved?-this would require that the two symmetries are in the same register across all motors.

– Structures of individual stator complexes are discussed in the section, "The stator-rotor interaction induces[…]", but are not depicted in any figures.

– How did the authors calculate stator occupancy? Stator occupancy determination (subsection “The conformational changes of the C-ring are directly linked to higher torque and faster motility”): "statistical analysis" needs description. At the moment the reader needs to dig into the supplemental figures to (partially) understand the procedure. This needs considerably greater description.

– There is a contradiction between the author's claim that the collar is likely "a scaffold for assembling and stabilizing the torque-generating stator units" and their observation that stator occupancy is a function of ion flux. The authors must clarify which they believe is stabilizing the stator units.

– The authors suggest that their results show that proton flux is required for stator binding to the motor, but they do no experiments where PMF is manipulated; rather, all their results show is that amino acid point mutations change stator occupancy, which is considerably less conclusive than their stated conclusions. To be able to claim that stator occupancy is proton flux dependent I would argue that additional experiments using CCCP to dissipate the PMF are needed.

– Subsection “CryoEM sample preparation”: more details on which grids used, approximate parameters of manual blotting, types of filter paper, etc.

– Subsection “Cryo-electron tomography”: post-processing: was data low-pass filtered? CTF corrected?

– How were stator rings and C-rings "stretched out": what software was used? How was the relative intensity plot (Figure 3—figure supplement 1) calculated?

– Figure 1: swarm plates very unclear and will need redoing.

– – Why are these data not CTF-corrected? There are large CTF artifacts which make some of the claims difficult to believe. At the very least, the authors should have truncated the data to the first zero of the CTF.

---

## [Author Response]

[Editors’ note: the author responses to the first round of peer review follow.]

The reviewers were all in agreement that the paper was very impressive from a technical point of view. However, the main unique conclusion of the paper, involving ion flow, was seen as speculative. It is quite possible that a substantially revised paper including actual data on ion flow would surmount the current concerns, but eLife does not offer such major revision decisions. If the authors can provide such a paper in the future we would be willing to consider it further.Reviewer #2:This interesting paper describes use of Borrelia burgdorferi as a model to study conformational changes during torque generation by the bacterial flagellar motor. The authors are particularly interested in the stator complex motor proteins, and image wild-type motors and compare them with two MotB mutants, D24E (reduced motility) and D24N (abolished motility), and a motB¬-deletion. Comparison of these structures reveals conformational changes in the C-ring-primarily a change in C-ring angle-that the authors suggest will provide clues to future understanding the mechanism of flagellar motor torque generation.The paper is well-written and technically well executed (indeed, is a technical tour-de-force in some ways), although has unclear biological significance. I have a number of comments:Major comments:While the results are noteworthy as technical accomplishments, it is unclear whether the "conformational change" is an important observation with implications for understanding torque generation, or whether it is simply that the C-ring is no longer held in place by the stator complexes. A possibly related observation of the C-ring conformation changing upon stator engagement has already been observed (in Campylobacter jejuni, Beeby and Hendrixson labs, Beeby et al., 2016), and interpreted as C-ring flexibility when not engaged by the stator complexes. Can the authors argue against this null hypothesis? If the authors do have reason to believe that this is a mechanistically-relevant conformation change as opposed to increased flexibility in the absence of the stator complexes, this would be a significant result. If not, the proposed "conformational changes" may not tell us much about torque generation.

We agree with the reviewer that the C-ring has its flexibility as suggested in *Campylobacter jejuni* (Beeby et al., 2016) and *Borrelia burgdorferi* (Qin et al., 2018). To reveal any conformational change with statistical significance, we analyzed over thousands of motors in each strain and then determined in situ motor structures at high resolution by subtomogram averaging. The C-ring structures are well resolved in the global averages, suggesting that the flexibility of the C-ring does not have severe impact on our in situ structural analysis and comparison. It is worthy to emphasize that the specific stator-induced conformational change of the C-ring has not been observed previously. To understand the mechanism underlying the conformational change, we specifically constructed a point mutant (motB-D24N), in which the MotB stator subunit is unable to conduct protons for torque generation. The conformation of the C-ring in motB-D24N is similar to that in Δ*motB*, suggesting that the torque generated by the stator is critical for the C-ring conformational change observed in the WT motor. Furthermore, we found that the C-ring has less conformational change in another point mutant (motBD24E), in which the MotB stator subunit is known to have lowered proton-conducting activity. Therefore, we strongly believe that the conformational change of the C-ring observed here is mechanistically relevant to the torque generated by the stator.

Use of the word "essential" in the title is inaccurate: nothing in the results reveals that these conformational changes are essential for flagellar rotation.

We changed the title to reflect the key message of the manuscript: “*Structural insights into* flagellar stator-rotor interactions”

The Materials and methods section, and descriptions of procedures used in the Results section, is entirely inadequate, and needs considerable elaboration, as follows.– How did the authors identify 46-fold symmetry? What software did the authors use? Was the C-ring averaged separately to the stator ring, and the two subsequently merged? If not, how were both symmetries resolved?-this would require that the two symmetries are in the same register across all motors.

We have provided detailed descriptions of the whole procedures from sample preparation to image analysis, although most information and details in the Materials and methods section were published previously (Liu et al., 2009; Zhao et al., 2013; Zhu et al., 2017). se Specifically, to determine the symmetry of the C-ring, we used sub-tomogram package i3 to do the alignment and classification as follows:

1) Do the asymmetric reconstruction for the whole motor and get the average motor structure as shown in Figure 2D in the manuscript. As the collar structure dominates the alignment, we are able to see the 16-fold symmetry of the stator ring and the collar (Figure 2E in the manuscript), but we could resolve the symmetry of the C-ring.

2) Then do focused alignment and classification on the C-ring. During the processing, we limited the angular search range between -2° and +2°.

3) After several cycles of refinement and 3D classification for the C-ring, we revealed the 46-fold symmetry of the C-ring in the WT motor. Importantly, after focused alignment and classification on the C-ring, the collar and the stator ring maintain16-fold symmetry.

– Structures of individual stator complexes are discussed in the section, "The stator-rotor interaction induces[…]", but are not depicted in any figures.

The structure of individual stator complex is shown in Figure 3I in the manuscript. The “individual” stator complex does not mean we exactly cut out one stator complex, it is just we focused at the region around one stator complex in order to get more structural details.

– How did the authors calculate stator occupancy? Stator occupancy determination (subsection “The conformational changes of the C-ring are directly linked to higher torque and faster motility”): "statistical analysis" needs description. At the moment the reader needs to dig into the supplemental figures to (partially) understand the procedure. This needs considerably greater description.

To estimate the stator occupancy, we took advantage of a unique spirochete-specific “collar” and the striking difference between averaged structures of the motors from WT and a *ΔmotB* mutant. Specifically, the unique collar structure with 16-fold symmetry is present in the motors from WT and the *ΔmotB* mutant. It provides 16 spots to interact with stator complexes. There is no stator in the Δ*motB* motor. In contrast, 16 stator complexes are visible in averaged structure of the WT motor. Because the collar feature provides well-defined spots for stator binding, we were able to analyze each of the 16 stator-corresponding spots from each motor after initial alignment of entire motors. Through focused alignment and classification, we generated class averages. Among the class averages, some have stator densities and others have no stator density. Therefore we were able to estimate the stator occupancy in each motor from several different strains. Importantly, our estimation from the Δ*motB* mutant and WT matches well with our visual observation from the motor average structures of the Δ*motB* mutant and WT.

– There is a contradiction between the author's claim that the collar is likely "a scaffold for assembling and stabilizing the torque-generating stator units" and their observation that stator occupancy is a function of ion flux. The authors must clarify which they believe is stabilizing the stator units.

We believe that both collar and ion flux are important for the assembly and function of the stator complexes. First, it has been well documented that ion flux is important for the assembly of the stator in *E. coli* and many other bacteria. We also showed in this manuscript that the stator occupancy reduced significantly in a point mutant motB-D24N in which the MotB stator subunit is unable conduct protons for torque generation. Second, the collar plays an important role in recruiting the stator complexes in *Borrelia*, which is supported by our recent study. Indeed, in several mutants lacking genes essential for the collar formation, both collar and stator are absent. Third, we showed in this study that stator complexes are still present in two point mutants (motB-D24N and motB-D24E). Together, our data suggested that the collar helps to recruit and stabilize stator complexes and ion flux is needed to generate the torque and to increase stator occupancy.

– The authors suggest that their results show that proton flux is required for stator binding to the motor, but they do no experiments where PMF is manipulated; rather, all their results show is that amino acid point mutations change stator occupancy, which is considerably less conclusive than their stated conclusions. To be able to claim that stator occupancy is proton flux dependent I would argue that additional experiments using CCCP to dissipate the PMF are needed.

MotB contains a highly conserved aspartic acid residue (Asp32 in *E. coli*, Asp24 in *Borrelia*). Extensive studies in *E. coli* and *Salmonella* indicated that it is located inside a proton channel and plays an essential role in proton transfer through the flagellar motor. Importantly, a conservative mutation D32N in *E. coli* abolished motor function, and another conservative mutation D32E retained reduced motor function. Here we showed the conservative mutations (D24E and D24N) in *Borrelia* have similar motility phenotypes as those in *E. coli*. In addition, we provided direct evidence that the mutations not only altered the stator occupancy but also reduced or even abolished the PMF-driven torque and the C-ring conformational change.

To further support our model, we did additional CCCP experiments as suggested by the reviewer. CCCP has been shown to rapidly cause immobilization of flagellar rotation in many bacteria including *B. burgdorferi* (Motaleb et al., 2000). CCCP-treated cells of wild-type *B. burgdorferi* were immobilized within 15 min (Motaleb et al., 2000). The paralyzed cells retained their flat-wave morphology. The averaged motor structure from the CCCP-treated cells is similar to that from motile WT cells. The stator complexes remain attached to the collar and the C-ring. However, the C-ring conformation in the CCCP-treated motor is different from that in motile WT motor. In contrast, the C-ring conformation in the CCCP-treated motor resembles those in the motB motor or the motBD24N motor. These results from CCCP-treated motor further support that PMF-driventorque induces C-ring conformational change required for flagellar rotation.

– Subsection “CryoEM sample preparation”: more details on which grids used, approximate parameters of manual blotting, types of filter paper, etc.

Our cryoEM sample preparation of *B. burgdorferi* has been described extensively in many papers (Liu et al., 2009; Zhao et al., 2013; Zhu et al., 2017). We have been consistently using the similar protocol. Nevertheless, we agree with the reviewer that more details should be included in the “Cryo-EM sample preparation” section.

– Subsection “Cryo-electron tomography”: post-processing: was data low-pass filtered? CTF corrected?

We did use low-pass filter (mtffilter in tomography package IMOD) to remove highfrequency noise. We also determined defocus and did CTF correction by using the “ctfphaseflip” function in IMOD. We add detailed information in the “Subtomogram averaging and correspondence analysis” section.

– How were stator rings and C-rings "stretched out": what software was used? How was the relative intensity plot (Figure 3—figure supplement 1) calculated?

UCSF Chimera was used to unroll the stator ring and the C-ring. The intensity plots shown in Figure 3—figure supplement 1 were measured by imageJ along the dashed lines shown in Figure 3—figure supplement 1 panels A and B. We have included more technical details in the “Subtomogram averaging and correspondence analysis” section

– Figure 1: swarm plates very unclear and will need redoing.

We did swarm plates again as suggested by the reviewer. We provide a new figure.

– Why are these data not CTF-corrected? There are large CTF artifacts which make some of the claims difficult to believe. At the very least, the authors should have truncated the data to the first zero of the CTF.

We used IMOD to determine the defocus and correct CTF in 2D, which is not as accurate as 3D CTF correction. Therefore, we did truncate our final averages at the resolution estimated by Fourier Shell Correlation coefficient.